# OpenReview forum: "VoxPrivacy: A Benchmark for Evaluating Interactional Privacy of Speech Language Models"
_ICLR.cc/2026/Conference — ICLR 2026 Poster_

### Official Review · Reviewer_G4Sy · 2025-10-28

**Soundness:** 2
**Presentation:** 3
**Contribution:** 2
**Rating:** 4
**Confidence:** 3

**Summary:**

VoxPrivacy is the first benchmark that evaluates interactional privacy for Speech Language Models (SLMs) in multi-user, spoken settings. It tests whether a model can keep user-specific secrets across three escalating tiers: (1) obeying explicit non-disclosure commands, (2) using the speaker’s voice as a key to disclose only to the original owner, and (3) proactively protecting privacy with no instruction by inferring sensitivity from content and context. Built from 7,107 utterances (32.86 hours, English/Chinese balanced) of high-quality synthetic audio with diverse speakers, VoxPrivacy pairs objective LLM-as-judge scoring with human validation. Across nine SLMs, most open-source systems hover around chance on conditional privacy decisions, revealing a core weakness in speaker-aware reasoning, not basic conversation. The authors also show a practical path forward: fine-tuning on a 4,000-hour training set substantially improves privacy compliance while preserving general abilities, though proactive, common-sense privacy remains challenging and vulnerable to spoofing attacks.

**Strengths:**

- Novel and interesting research questions
- Detailed and complete experiment
- Well-written paper

**Weaknesses:**

- Lack of practical motivations
- Data authenticity and generalization issues

**Questions:**

This paper presents VoxPrivacy, the first benchmark designed to evaluate interactional privacy for speech language models (SLMs) in realistic multi-user spoken scenarios. The authors construct a bilingual (English/Chinese) dataset containing 7,107 utterances (32.86 hours) of high-quality synthetic audio from diverse speakers and evaluate nine SLMs, including both open-source and proprietary systems. The benchmark employs a hybrid evaluation protocol combining LLM-as-judge scoring and human validation. Results show that open-source SLMs perform near random on speaker-conditioned privacy tasks, while closed-source models and fine-tuned versions achieve better compliance. Additional analyses reveal that the main bottleneck lies not in conversational ability but in speaker-aware reasoning and contextual privacy understanding. The paper is well-written in general. However, I do have the following concerns:

- Lack of practical motivation.
Although it sounds fancy to leverage models' internal ability to conduct the permission verification, it is still common sense that models' responses are unreliable and random. Therefore, it may be possible to construct the permission systems instead of relied solely on large models to solve the questions. The authors also show in the paper that some jailbreak method can break the model recognition ability. Therefore, I am wondering given the current model architecture and model design, is it worth doing such a test? Should all permission-related issues be handed over to a dedicated permission system?

- Data authenticity and generalization issues
All speech data in VoxPrivacy are synthetic. WHile this choice ensures ethical safety, it limits the acoudstic and semantic diversity presnet in real spoken interactions, such as bacground noise, accent variation, and spontaneous interruptions. As a result, the benchmark may overestimate the model performance on the more realisitic data.

---

> ### Author Response · Authors · 2025-11-21
> **Reply to Reviewer G4Sy (Part 1/2)**
>
> We sincerely thank you for the thoughtful questions. We particularly appreciate the challenge regarding the practical motivation, as it allows us to clarify why interactional privacy is a distinct problem that cannot be solved by traditional permission systems. We have also conducted new experiments with real human speech to address your concerns about data authenticity.
>
> ### **(1) Practical Motivation: Why External Permission Systems Are Insufficient (Response to Question 1)**
>
> >“Weakness: Lack of practical motivation... it may be possible to construct the permission systems instead of relied solely on large models... Therefore, I am wondering given the current model architecture and model design, is it worth doing such a test? Should all permission-related issues be handed over to a dedicated permission system?”
>
> `Please refer to Line 78-81 on Page 2 of the revised manuscript.`
>
> We agree that today many permission checks are done outside the model (accounts, device locks, Siri-style voice match) and that current SLMs are unreliable and jailbreakable. However, we argue that in the **shared, multi-user environments** where SLMs are increasingly deployed, such as smart homes, in-car assistants, and group meeting tools, external permission systems alone are fundamentally inadequate, not merely insufficient.
>
> The core issue is that external permission systems require **pre-enrollment** and **user-authentication**. In a shared environment like a family smart speaker or a collaborative workspace assistant, this approach breaks down for two critical reasons:
>
> - First, it is impractical to require every person who might interact with the assistant to pre-register their voice profile, especially for guests, temporary users, or ad-hoc participants in group conversations.
> - Second, and more fundamentally, authenticating users through an external system would create a "hard isolation" between users' conversational contexts. This means User A's statements would be strictly inaccessible when processing User B's query, even when collaborative information sharing is appropriate (e.g., "What time did Alex say the meeting starts?"). Such rigid segregation directly contradicts the collaborative nature of shared assistants and makes natural multi-user dialogue impossible.
>
> **Our position is therefore that interactional privacy must be handled internally by the SLM precisely because the assistant needs access to the full conversational history across speakers to function naturally in shared settings.** The model must simultaneously see what different users have said (to enable collaboration) while using speaker identity, conversational context, and content sensitivity to decide what to reveal to whom. This is not a choice between internal reasoning and external systems—it is a recognition that the shared-environment use case requires capabilities that external authentication architectures cannot provide. External systems may still control device access at a coarse level (e.g., whether someone can use the device at all), but they cannot solve the fine-grained, context-dependent privacy decisions that arise **within** an ongoing multi-user conversation.
>
> We believe it is worthwhile to test this behavior even with current architectures because our results show that, under clean and controlled conditions, existing SLMs fail to respect basic speaker-specific privacy rules. This evidence is critical for guiding future designs: it demonstrates that the problem is not hypothetical but present and measurable, and that solutions must go beyond architectural improvements to include better training data and reasoning capabilities for shared conversational contexts.

---

> ### Author Response · Authors · 2025-11-21
> **Reply to Reviewer G4Sy (Part 2/2)**
>
> ### **(2) Data Authenticity: Validation with Real Human Speech (Response to Question 2)**
>
> >“Weakness: Data authenticity and generalization issues... All speech data in VoxPrivacy are synthetic. While this choice ensures ethical safety, it limits the acoustic and semantic diversity present in real spoken interactions... As a result, the benchmark may overestimate the model performance on the more realistic data.”
>
> `Please refer to Line 404-428 on Page 8, Line 1500-1511 on Page 28 of the revised manuscript.`
>
> We agree that synthetic audio lacks the complexity of real speech (background noise, stuttering). To determine if our synthetic benchmark is a valid proxy for real-world performance, we conducted a **"Real-VoxPrivacy" validation study** during the rebuttal period.
>
> - **Setup:** We recruited **18 bilingual participants** to record a subset of our test scripts, resulting in **586 real human utterances** (288 English, 298 Chinese). We evaluated our baseline models on this real-world audio.
> - **Findings (Table 4 of the revised manuscript):**
>     1. **Consistent Rankings:** The relative performance of models remained consistent. For example, Gemini-2.5-Pro remained the top performer in Tier 1, and open-source models continued to struggle near random chance on Tiers 2/3.
>     2. **Validation of Difficulty:** The performance drop from Tier 2 (Instruction) to Tier 3 (Inference) was preserved in the real audio, confirming this is a cognitive failure, not an artifact of synthesis.
>     3. **"Best-Case" Baseline:** Interestingly, some models performed *worse* on real audio due to accents and noise (as you predicted). This reinforces the value of VoxPrivacy: if models fail on our clean, synthetic "best-case" data, they are guaranteed to fail in the wild. Our benchmark establishes the **necessary upper bound** for model capabilities.
>
> **Table 4 of the revised manuscript. Validation on Real Speech: Model performance on the human-recorded Real-VoxPrivacy dataset.**
>
> | Model | T1-EN-IRR↓ | T1-EN-Acc↑ | T1-ZH-IRR↓ | T1-ZH-Acc↑ | T2-EN-Acc↑ | T2-EN-F1↑ | T2-ZH-Acc↑ | T2-ZH-F1↑ | T3-EN-Acc↑ | T3-EN-F1↑ | T3-ZH-Acc↑ | T3-ZH-F1↑ |
> |:---|:---:|:---:|:---:|:---:|:---:|:---:|:---:|:---:|:---:|:---:|:---:|:---:|
> |_**Upper Bound**_ | | | | |  | | | | | | | |
> | LLM | 0.69 | 97.57 | 0.34 | 98.83 | 92.36 | 90.31 | 91.95 | 91.52 | 86.11 | 85.03 | 85.91 | 85.10 |
> |_**Closed-sourced**_| | | | |  | | | | | | | |
> | Gemini-2.0-flash | 0.79 | 72.44 | **0.40** | _86.35_ | 60.10 | 59.30 | 63.84 | 47.74 | 61.16 | 47.78 | 56.39 | 36.13 |
> | Gemini-2.5-pro | _0.39_ | **90.83** | 1.36 | **88.18** | _74.92_ | _71.71_ | _78.19_ | _73.11_ | _71.20_ | _62.89_ | _76.81_ | **70.17** |
> |_**Open-sourced**_  | | | | | | | | | | | | |
> | Qwen2.5Omni | **0.00** | 31.73 | _0.67_ | 24.41 | 50.88 | 34.12 | 49.83 | 21.16 | 51.96 | 49.81 | 50.17 | 19.67 |
> | MiniCPM-o2.6 | 1.04 | 16.67 | _0.67_ | 15.77 | 53.57 | 33.67 | 49.83 | 26.60 | 53.33 | 32.49 | 48.82 | 12.64 |
> | Kimi-Audio | 8.68 | 63.96 | 7.03 | 34.52 | 48.89 | 49.26 | 51.24 | 34.91 | 51.96 | 49.81 | 49.82 | 19.21 |
> | Ours: Kimi-Audio-sft | 7.29 | _82.36_ | 10.40 | _80.54_ | **87.68** | **86.54** | **85.63** | **84.71** | **80.32** | **79.81** | **77.11** | _68.29_ |
>
> >_Notes:_ T1/T2/T3 = Tier 1/2/3; Acc = Accuracy; **Bold** = best; _Italic_ = second best; ↑/↓ = higher/lower is better.
>
> We again thank you for these important points; they help us better explain both the practical role of interactional privacy and the current limits of our synthetic-audio design.

---

### Official Review · Reviewer_KeNS · 2025-10-31

**Soundness:** 4
**Presentation:** 4
**Contribution:** 4
**Rating:** 10
**Confidence:** 5

**Summary:**

Privacy studies have, up to now, mainly focused on individual users, whereas many problems of privacy occur only within the context of an interaction. This paper proposes a new dataset for evaluating interactional privacy when interacting with speech agents. This is an important contribution and a problem that has been overlooked in past works.
The dataset is synthetic speech audio. This is a reasonable starting point, though obviously, actual recordings of human speech would be better. Recording sufficient amounts of such data, however, requires significant resources. Synthetically created speech is thus the obvious and reasonable simplification.
Overall, I like the paper rather a lot and have only minor comments.

**Strengths:**

- High novelty; The addressed research problem is novel, and this is, as far as I know, the first dataset and methodology for evaluating interactional privacy.
- High quality; The proposed dataset is designed following principles of good design, the validation tests for the dataset are good, and the analysis of results is insightful.
- Good clarity; Writing and argumentation are clear, with only minor blemishes.
- High significance: As this work addresses an important problem that has not been studied before, I believe that this can have a significant impact.

**Weaknesses:**

Main weaknesses:
- Argumentation: Building a dataset for SLMs was motivated by the fact that spoken dialogues have plenty of contextual information that is not available in the text only. This is true; speech is a much more informative representation than text, and my informed guess is that much of the information related to interactional privacy is available only in the voice (not in text). That said, as data is here created through synthesis from text, there is no way to confirm that information related to interactional privacy (beyond text) is included in the dataset. The question is thus whether the audio representation has any added value in comparison to text only, as long as the data is synthetic? I acknowledge that this is a difficult question that probably cannot be solved in a single paper and probably not even in a single doctoral thesis, but I would request a discussion about this issue in the paper.
- Data representativeness: A variation of the above argument is that synthetic data is always a proxy for real data, and special care must be taken to ensure or verify that it represents the true population adequately. This could be solved, for example, by adding audio from real human speakers to the test set. If the performance with synthetic and authentic samples are similar, then the synthetic data is sufficient. Again, this is not a demand but a proposal, given that it this modification can require significant resources, time and effort.

Minor comments:
- Fig 2, stage 3; spelling of "Instruction"
- Fig 2 is very dense, packed with information. I don't have a better solution to propose, but I just want to acknowledge that understanding the figure requires some effort.

**Questions:**

My suggestions and questions were included in the weaknesses box.

---

> ### Author Response · Authors · 2025-11-21
> **Reply to Reviewer KeNS (Part 1/2)**
>
> We are sincerely grateful for your exceptional rating (10/10) and your recognition of this work as "novel", "high quality" and "significant". Your constructive comments regarding the role of audio and data representativeness have helped us refine the core arguments of our paper.
>
> ### **(1) The Value of Audio in Synthetic Data (Response to Weakness 1)**
>
> >“Weakness: The question is thus whether the audio representation has any added value in comparison to text only, as long as the data is synthetic? ... I would request a discussion about this issue in the paper.”
>
> `Please refer to Line 534-539 on Page 10 of the revised manuscript.`
>
> You raise a profound question: *If the audio is synthesized from text, does it add value beyond the text itself?*
>
> We argue that the audio representation adds a critical dimension that text cannot capture, even when synthetic: **Biometric-Semantic Integration.**
>
> - **In Text-only privacy (The Upper Bound):** The speaker identity is provided as an explicit, infallible label (e.g., `<Speaker 1>: Keep this secret`). The model only needs to solve the logical constraint.
> - **In Speech (VoxPrivacy):** The model must extract the identity *from the acoustic waveform* and bind it to the semantic instruction. The "added value" of our audio, even though synthetic, is that it forces the model to perform **implicit speaker verification**.
> - **Evidence:** Our results show a massive gap between the Text-Only Upper Bound (around 90% accuracy) and SLMs on Audio (around 50% accuracy). This proves that the failure mode is not in the logic (which is in the text) but in the **inability to map acoustic identity to privacy rules**. Furthermore, our **Spoofing Attack** (Table 5) specifically exploits the acoustic dimension (timbre similarity), a threat vector that simply does not exist in text.
> - **Revision:** we expand the discussion in Section 7 of our revised manuscript to clarify that while synthetic audio may lack paralinguistic nuance like emotion, it successfully isolates the core *biometric* challenge of interactional privacy.
>
> ### **(2) Data Representativeness & Real Recordings (Response to Weakness 2)**
>
> >“Weakness: Special care must be taken to ensure or verify that it represents the true population adequately. This could be solved, for example, by adding audio from real human speakers to the test set.”
>
> `Please refer to Line 404-428 on Page 8, Line 1500-1511 on Page 28 of the revised manuscript.`
>
> We fully thank your suggestion to verify our synthetic data against real human speech. Therefore, we conducted a **"Real-VoxPrivacy" validation study**:
>
> - **Setup:** We recruited **18 bilingual volunteers** to record a representative subset of our scripts, resulting in **586 authentic utterances** (288 English, 298 Chinese). We evaluated our baseline models on this real-world set.
> - **Findings:** The results strongly validate the synthetic benchmark **as shown in Table 4 of the revised manuscript**.
>     1. **Consistency:** The relative ranking of models remained consistent (e.g., Gemini-2.5-Pro outperformed open-source models).
>     2. **Difficulty Curve:** The performance drop from Tier 2 (Instruction) to Tier 3 (Inference) observed in synthetic data was preserved in real data.
>     3. **Validation:** This confirms that the "random chance" performance of open-source models is a fundamental model weakness, not an artifact of synthetic TTS audio.
>
> **Table 4 of the revised manuscript. Validation on Real Speech: Model performance on the human-recorded Real-VoxPrivacy dataset.**
>
> | Model | T1-EN-IRR↓ | T1-EN-Acc↑ | T1-ZH-IRR↓ | T1-ZH-Acc↑ | T2-EN-Acc↑ | T2-EN-F1↑ | T2-ZH-Acc↑ | T2-ZH-F1↑ | T3-EN-Acc↑ | T3-EN-F1↑ | T3-ZH-Acc↑ | T3-ZH-F1↑ |
> |:---|:---:|:---:|:---:|:---:|:---:|:---:|:---:|:---:|:---:|:---:|:---:|:---:|
> |_**Upper Bound**_ | | | | |  | | | | | | | |
> | LLM | 0.69 | 97.57 | 0.34 | 98.83 | 92.36 | 90.31 | 91.95 | 91.52 | 86.11 | 85.03 | 85.91 | 85.10 |
> |_**Closed-sourced**_| | | | |  | | | | | | | |
> | Gemini-2.0-flash | 0.79 | 72.44 | **0.40** | _86.35_ | 60.10 | 59.30 | 63.84 | 47.74 | 61.16 | 47.78 | 56.39 | 36.13 |
> | Gemini-2.5-pro | _0.39_ | **90.83** | 1.36 | **88.18** | _74.92_ | _71.71_ | _78.19_ | _73.11_ | _71.20_ | _62.89_ | _76.81_ | **70.17** |
> |_**Open-sourced**_  | | | | | | | | | | | | |
> | Qwen2.5Omni | **0.00** | 31.73 | _0.67_ | 24.41 | 50.88 | 34.12 | 49.83 | 21.16 | 51.96 | 49.81 | 50.17 | 19.67 |
> | MiniCPM-o2.6 | 1.04 | 16.67 | _0.67_ | 15.77 | 53.57 | 33.67 | 49.83 | 26.60 | 53.33 | 32.49 | 48.82 | 12.64 |
> | Kimi-Audio | 8.68 | 63.96 | 7.03 | 34.52 | 48.89 | 49.26 | 51.24 | 34.91 | 51.96 | 49.81 | 49.82 | 19.21 |
> | Ours: Kimi-Audio-sft | 7.29 | _82.36_ | 10.40 | _80.54_ | **87.68** | **86.54** | **85.63** | **84.71** | **80.32** | **79.81** | **77.11** | _68.29_ |
>
> >_Notes:_ T1/T2/T3 = Tier 1/2/3; Acc = Accuracy; **Bold** = best; _Italic_ = second best; ↑/↓ = higher/lower is better.

---

> ### Author Response · Authors · 2025-11-23
> **Reply to Reviewer KeNS (Part 2/2)**
>
> ### **(3) Minor Comments**
>
> >“Minor comments: Fig 2, stage 3; spelling of "Instruction"... Fig 2 is very dense... understanding the figure requires some effort.”
>
> `Please refer to Line 184-187 on Page 4 of the revised manuscript.`
>
> - **Figure 2:** We have corrected the spelling of "Instruction".
> - **Figure Density:** We agree Figure 2 is dense. We revise the caption to include a "step-by-step reading guide" to help readers understand the data generation pipeline more easily.
>
> Thank you again for championing this work. We believe these revisions will substantially strengthen the paper's contribution.

---

> ### Comment · Reviewer_KeNS · 2025-11-25
> **Response to review-comments**
>
> Thank you for the updates and especially the new subjective testing. My questions have been satisfied and I keep my final assessment of scoring.

---

> > ### Author Response · Authors · 2025-11-25
> >
> > Dear reviewer KeNS,
> >
> > Thank you for taking the time to review our rebuttal. We are glad to have addressed your concerns! Once again, we appreciate your detailed and helpful review, as well as your positive feedback.

---

### Official Review · Reviewer_muR9 · 2025-11-01

**Soundness:** 2
**Presentation:** 2
**Contribution:** 2
**Rating:** 4
**Confidence:** 4

**Summary:**

This paper presents VoxPrivacy, a benchmark for evaluating a privacy failure mode (termed "interactional privacy") in multi-user conversational context for speech language models. One user discloses secret information to the SLM, explicitly or implicitly, and the SLM needs to ensure this secret information is not disclosed when another user queries it. VoxPrivacy consists of three tiers of privacy evaluation, ranging from the most explicit (directly indicating secrecy) to the most implicit (inferring protection needs based on commonsense). Their evaluation found current SLMs, especially open-source ones, struggling with these tasks. Their fine-tuned model improved the privacy preservation capabilities without compromising the general capabilities.

**Strengths:**

- The paper examines contextual privacy leakage issues in speech language models and engages with the unique capabilities of SLM to process the voice which can uniquely identify a person. Hence, it makes sense to evaluate the end-to-end privacy protection for SLM.
- The paper develops a benchmark covering both direct and indirect indicators of privacy information to perform a thorough evaluation of the privacy protection capabilities of closed-source and open-source models.
- The evaluation reveals substantial gaps in the open-source models' privacy preservation capabilities.
- The fine-tuned model showed promising results improving the privacy capabilities while preserving the general capabilities.

**Weaknesses:**

- I can't find a realistic grounding for the privacy violations in the benchmark. The benchmark assembles the specification, instruction, and probing queries into a multi-turn dialogue, which corresponds to the situation when multiple users converse with the SLM in the same session. In these cases, people already have equal access to the output of the model, which means the sensitivity of information in the output should be determined by everyone present in the conversation, rather than just the speaker (or even people who are co-present in the context with access to the model output, regardless of whether they participate in the conversation). Also, in this case it's not natural for one speaker to describe their secret in front of another. This scenario hence feels contrived.
- In the Tier 3 example, I don't understand why "I'm worried about my medical results" implies that "medical results" should be considered a secret. I feel the bar might be set too rigidly and doesn't align with common sense.
- Gaps are exaggerated in the abstract "most models perform near random chance, about 50% accuracy on binary privacy decisions" — In fact, the closed models performed well in many of the tier 1 and 2 tasks, sometimes even better than the fine-tuned model.
- I feel it's inappropriate to call the LLM-as-a-Judge evaluation as objective evaluation and the human evaluation as subjective, because they are following the same criteria.

**Questions:**

- Can you discuss the validity of your threat model, and why the benchmark design appropriately reflects it?
- Can you explain how you determine the test cases in Tier 3, specifically what procedure did you follow to ensure they properly reflect social norms and commonsense?
- Are there any considerations about the potentially different cultural norms between English vs. Chinese speaking contexts?
- How were the human annotations used to validate the LLM-as-a-Judge labels?

---

> ### Author Response · Authors · 2025-11-21
> **Reply to Reviewer muR9 (Part 1/2)**
>
> We sincerely thank you for the detailed assessment. We value your feedback on the threat model and scenario design, which we have addressed below. We hope these clarifications and our new human validation results demonstrate the robustness of our benchmark.
>
> ### **(1) Threat Model Realism: Asynchronous Memory, Not Concurrent Chat (Response to Weakness 1 & Question 1)**
>
> >“Weakness: I can't find a realistic grounding for the privacy violations... multi-turn dialogue... corresponds to the situation when multiple users converse with the SLM in the same session... This scenario hence feels contrived.”
>
> >“Question: Can you discuss the validity of your threat model, and why the benchmark design appropriately reflects it?”
>
> `Please refer to Line 51-73 on Page 1 & 2, Line 192-194 on Page 4 of the revised manuscript.`
>
> We appreciate the opportunity to clarify a critical misunderstanding. **We fully agree with the reviewer that concurrent multi-user conversation is a vital use case for SLMs in shared environments.** In such real-time scenarios, as you noted, privacy logic relies on complex social dynamics between present parties. However, VoxPrivacy specifically targets a complementary but equally critical risk: **Asynchronous, Time-Separated Interactional Privacy**. This scenario simulates a shared device (e.g., a smart home speaker or in-car assistant) with long-term memory used by a family over times. The "3-turn" structure in our benchmark represents a compressed timeline of distinct events:
>
> 1. **Time $T_0$ (Former Session):** User A interacts with the device alone and discloses information (e.g., *“I’m worried about my medical results.”*).
> 2. **Time $T_1$ (Instruction):** User A sets a privacy preference (explicitly or implicitly).
> 3. **Time $T_2$ (Later Session):** Later, User B interacts with the same device. Because modern SLMs have long-term memory, User B asks a question that might trigger the retrieval of User A’s data.
>
> The "threat" is that the SLM acts as a naive information bridge between these separate sessions. This is a major, unsolved hurdle for deploying long-term memory in shared spaces. We restate this part in the Introduction and Task Definition of the revised manuscript.
>
> ### **(2) Tier 3 Validity & Sensitivity Standards (Response to Weakness 2 & Question 2)**
>
> >“Weakness: I don't understand why "I'm worried about my medical results" implies that "medical results" should be considered a secret. I feel the bar might be set too rigidly...”
>
> >“Question: Can you explain how you determine the test cases in Tier 3... to ensure they properly reflect social norms and commonsense?”
>
> `Please refer to Line 260-263 on Page 5 of the revised manuscript.`
>
> You questioned why "I'm worried about my medical results" implies a secret. We justify this through two lenses:
>
> - **Empirical Human Validation (New Experiment):** we conducted a user study with **5 bilingual annotators** who rated 200 randomly sampled secrets (including Tier 3 examples) on a Likert scale (1=Not sensitive, 5=Very sensitive).
>     - **Result:** **92% of the samples received an average rating of ≥4.** This confirms that human annotators overwhelmingly agree that statements like the medical example (which implies the existence of a private health condition) are sensitive and should not be shared with others in a household without consent.
> - **Safety-First Design:** We designed secrets to align with established regulatory frameworks like the General Data Protection Regulation (GDPR) (which classifies "data concerning health" as a special category) and the Health Insurance Portability and Accountability Act (HIPAA). In a deployed safety system, the cost of a False Negative (leaking private health anxiety to a third party) is significantly higher than a False Positive (refusing to answer).
>
> ### **(3) Refining Abstract Claims (Response to Weakness 3)**
>
> >“Weakness: Gaps are exaggerated in the abstract "most models perform near random chance..." — In fact, the closed models performed well...”
>
> `Please refer to Line 26-28 on Page 1 of the revised manuscript.`
>
> We agree that our original summary lumped all models together. We will revise the abstract to be more precise:
>
> *"most open-source models perform close to random chance (around 50% accuracy) on conditional privacy decisions, while even strong closed-source systems still fall short on proactive privacy inference."*

---

> ### Author Response · Authors · 2025-11-21
> **Reply to Reviewer muR9 (Part 2/2)**
>
> ### **(4) Evaluation Terminology & Validation (Response to Weakness 4 & Question 4)**
>
> >“Weakness: I feel it's inappropriate to call the LLM-as-a-Judge evaluation as objective evaluation and the human evaluation as subjective...”
>
> >“Question: How were the human annotations used to validate the LLM-as-a-Judge labels?”
>
> `Please refer to Line 300, 309 on Page 6 of the revised manuscript.`
>
> We agree that "objective/subjective" was imprecise. We will rename these to **"LLM-based Evaluation"** and **"Human Evaluation"**.
>
> Regarding validation (detailed in Appendix A.8): Human annotations were used as the **ground truth** to validate the LLM Judge. We sampled 800 dialogues and had 3 annotators rate them using the exact same guidelines as the LLM.
>
> - **Agreement:** We achieved a **Fleiss’ Kappa of 0.92**, indicating near-perfect agreement among humans.
> - **Correlation:** The Human-scored accuracy tracked the LLM-scored accuracy within a small margin (detailed in Table 2), confirming the automatic metric is a reliable proxy for human judgment.
>
> ### **(5) Cultural Norms (Response to Question 3)**
>
> >“Question: Are there any considerations about the potentially different cultural norms between English vs. Chinese speaking contexts?”
>
> `Please refer to Line 530-532 on Page 10 of the revised manuscript.`
>
> We designed the benchmark around universally sensitive categories (Health, Finance, Illicit Acts). However, we acknowledge that the threshold for privacy varies by culture (e.g., "Academic Performance" might be more sensitive in some cultures than others). While our bilingual validation confirms the current dataset is valid for both EN/ZH, we add a limitation in Section 7 of the revised manuscript noting that future work should explore culture-specific privacy norms (e.g., "collectivist vs. individualist" privacy definitions).
>
> We believe these revisions will substantially strengthen the paper's clarity and grounding. We appreciate the reviewer's detailed engagement and would welcome further discussion during the rebuttal period.

---

### Official Review · Reviewer_m294 · 2025-11-10

**Soundness:** 3
**Presentation:** 4
**Contribution:** 3
**Rating:** 8
**Confidence:** 3

**Summary:**

This paper introduces VoxPrivacy, the first benchmark for assessment of interactional privacy in SLMs operating in shared, multi-user environments. The authors propose an evaluation framework that tests (1) direct command secrecy, (2) speaker-verified access, and (3) proactive privacy — reflecting practical privacy expectations in systems like smart home assistants. The benchmark consists of 7107 synthetic utterances (32.86 hours, English and Chinese). It covers a variety of privacy-sensitive categories constructed with a multi-stage pipeline and verified for quality and linguistic diversity. 9  SLMs are evaluated, showing that most open-source models perform no better than random chance on speaker-aware privacy tasks. Closed-source and fine-tuned models perform better but still show significant vulnerabilities. Analysis highlights crucial challenges in contextual integration and adversarial robustness. The authors release all resources, including the benchmark, a 4000-hour mixed-task training set, and a baseline model.

**Strengths:**

1. The paper addresses a novel and important problem in SLMs, interactional privacy in multi-user environments, which is underexplored. The introduction of the VoxPrivacy benchmark, based on a theoretically grounded definition of interactional privacy using Nissenbaum's Contextual Integrity.

2. The quality of the paper is good, the authors constructed a large-scale bilingual dataset with data synthesis, filtering, multi-model LLM generation, and human verification processes. The benchmark includes well-designed three-tiered tasks isolating distinct privacy capabilities, with comprehensive evaluations of nine state-of-the-art models, including open- and closed-source systems.

3. The paper is very well written.

4. The experimental results, based on a carefully constructed benchmark and clear evaluation methods, support the claim that current SLMs (especially open-source) struggle to reliably enforce speaker-based privacy.

5. The benchmark addresses critical safety and privacy challenges faced by SLMs in realistic shared environments such as smart homes. This benchmark will foster further research and development of practical solutions for privacy-preserving SLMs.

**Weaknesses:**

1. Synthetic dataset limitations (also acknowledged by the authors). The use of only synthetic, LLM-generated dialogues for privacy-sensitive utterances may reduce real-world relevance. The paper lacks user studies or comparisons with real data to confirm if synthetic secrets match actual privacy concerns.

2. Artificial dialogue structure.The fixed 3-turn dialogue pattern (secret statement → privacy instruction → probe) may not fully capture the richness and variability of natural conversations, including interruptions, multi-party interplay, and temporal gaps.
Speaker verification analysis and metric. Some details of speaker verification analysis are missing, also more conventional automatic speaker verification metric, i.e. equal error rate,  would be more appropriate.

3. Limited fine-tuning analysis. The construction of the 4000-hour mixed-task dataset for fine-tuning, including mixtures of tasks and proportions, is not fully justified. Ablation studies exploring the impact of different auxiliary tasks and the balance between privacy enhancement and general capability preservation are missing.

4. Cross-lingual performance gaps. The underperformance on Chinese w.r.t. English is observed but is not sufficiently analyzed, leaving open questions about multilingual robustness.

**Questions:**

1. Have the authors conducted user studies to validate that humans perceive the synthetic secrets as privacy-sensitive and expect SLMs to protect them?

2. Can the authors provide error analyses by secret categories and instruction phrasing to clarify which types of secrets leak most frequently?

3. Can the authors elaborate on the acoustic features or embeddings models use for speaker verification and their limitations, particularly regarding spoofing attacks?

4. Can the authors discuss potential extensions of the benchmark to more realistic, interactive multi-turn dialogues?
Regarding the multilingual aspect, do the authors have hypotheses or insights on why models underperform for Chinese?

---

> ### Author Response · Authors · 2025-11-21
> **Reply to Reviewer m294 (Part 1/3)**
>
> We sincerely thank you for your positive assessment and constructive feedback. Your recognition of our work's novelty, technical quality, and significance is greatly appreciated. We have conducted additional experiments to directly address your concerns.
>
> ### **(1) Validation with Real-World Data (Response to Weakness 1 & Question 1)**
>
> >“Weakness: The use of only synthetic, LLM-generated dialogues for privacy-sensitive utterances may reduce real-world relevance. The paper lacks user studies or comparisons with real data to confirm if synthetic secrets match actual privacy concerns.”
>
> >“Question: Have the authors conducted user studies to validate that humans perceive the synthetic secrets as privacy-sensitive and expect SLMs to protect them?”
>
> `Please refer to Line 259-263 on Page 5, Line 404-428 on Page 8, Line 1500-1511 on Page 28 of the revised manuscript.`
>
> We agree that validating synthetic benchmarks against real-world data is crucial. We have performed two studies to confirm the validity of VoxPrivacy:
>
> - **Human Perception Study:** To verify that our synthetic secrets align with human privacy norms, we asked 5 bilingual participants to rate the sensitivity of 200 randomly sampled secrets on a Likert scale (1=Not sensitive, 5=Very sensitive). **92% of the secrets received an average rating of ≥4**, confirming that humans perceive the content as private and worthy of protection.
> - **Validation on Real Human Speech:** To address the concern about synthetic audio, we recruited **18 bilingual participants** to record a "Real-VoxPrivacy" test set, consisting of **586 authentic utterances** (288 English, 298 Chinese). We evaluated our baselines on this real-world set.
>     - **Results:** **The results are presented in Table 4 of the revised manuscript.** We observe a strong alignment between model performance on synthetic audio and real human speech. For instance, in the Tier 1 task, the performance ranking of models remain consistent (e.g., Gemini-2.5-Pro remained the top performer). The performance gap between Tier 2 and Tier 3 also persist in the real data, confirming that the challenges revealed by VoxPrivacy are inherent to the models, not artifacts of synthesis.
>
> **Table 4 of the revised manuscript. Validation on Real Speech: Model performance on the human-recorded Real-VoxPrivacy dataset.**
>
> | Model | T1-EN-IRR↓ | T1-EN-Acc↑ | T1-ZH-IRR↓ | T1-ZH-Acc↑ | T2-EN-Acc↑ | T2-EN-F1↑ | T2-ZH-Acc↑ | T2-ZH-F1↑ | T3-EN-Acc↑ | T3-EN-F1↑ | T3-ZH-Acc↑ | T3-ZH-F1↑ |
> |:---|:---:|:---:|:---:|:---:|:---:|:---:|:---:|:---:|:---:|:---:|:---:|:---:|
> |_**Upper Bound**_ | | | | |  | | | | | | | |
> | LLM | 0.69 | 97.57 | 0.34 | 98.83 | 92.36 | 90.31 | 91.95 | 91.52 | 86.11 | 85.03 | 85.91 | 85.10 |
> |_**Closed-sourced**_| | | | |  | | | | | | | |
> | Gemini-2.0-flash | 0.79 | 72.44 | **0.40** | _86.35_ | 60.10 | 59.30 | 63.84 | 47.74 | 61.16 | 47.78 | 56.39 | 36.13 |
> | Gemini-2.5-pro | _0.39_ | **90.83** | 1.36 | **88.18** | _74.92_ | _71.71_ | _78.19_ | _73.11_ | _71.20_ | _62.89_ | _76.81_ | **70.17** |
> |_**Open-sourced**_  | | | | | | | | | | | | |
> | Qwen2.5Omni | **0.00** | 31.73 | _0.67_ | 24.41 | 50.88 | 34.12 | 49.83 | 21.16 | 51.96 | 49.81 | 50.17 | 19.67 |
> | MiniCPM-o2.6 | 1.04 | 16.67 | _0.67_ | 15.77 | 53.57 | 33.67 | 49.83 | 26.60 | 53.33 | 32.49 | 48.82 | 12.64 |
> | Kimi-Audio | 8.68 | 63.96 | 7.03 | 34.52 | 48.89 | 49.26 | 51.24 | 34.91 | 51.96 | 49.81 | 49.82 | 19.21 |
> | Ours: Kimi-Audio-sft | 7.29 | _82.36_ | 10.40 | _80.54_ | **87.68** | **86.54** | **85.63** | **84.71** | **80.32** | **79.81** | **77.11** | _68.29_ |
>
> >_Notes:_ T1/T2/T3 = Tier 1/2/3; Acc = Accuracy; **Bold** = best; _Italic_ = second best; ↑/↓ = higher/lower is better.

---

> ### Author Response · Authors · 2025-11-21
> **Reply to Reviewer m294 (Part 2/3)**
>
> ### **(2) Dialogue Structure and Speaker Verification (Response to Weakness 2, Questions 3 & 4)**
>
> >“Weakness: The fixed 3-turn dialogue pattern... may not fully capture the richness and variability of natural conversations... Some details of speaker verification analysis are missing, also more conventional automatic speaker verification metric, i.e. equal error rate, would be more appropriate.”
>
> >“Question: Can the authors elaborate on the acoustic features or embeddings models use for speaker verification...?”
>
> >“Question: Can the authors discuss potential extensions of the benchmark to more realistic, interactive multi-turn dialogues?”
>
> `Please refer to Line 1321-1345 on Page 25 of the revised manuscript.`
>
> - **Dialogue Structure:** While we use a canonical `secret → instruction → probe` structure for controlled evaluation, we agree that natural conversation is messier. We discuss this via the **Needle-in-the-Haystack Test (Section 5.3, Appendix A.5)**, which inserts irrelevant turns between the instruction and the probe, simulating natural dialogue. The performance drops observed there confirm that models struggle with privacy in extended, naturalistic contexts.
> - **Speaker Verification (SV) Metrics:** As suggested, we calculated the **Equal Error Rate (EER)** to complement the Accuracy metric in Table 13 of the revised manuscript. As shown below, EER shows a strong negative correlation with accuracy, offering a conventional view of the models' biometric capabilities.
> - **Acoustic Features:** Regarding your question on features: unlike modular systems that use dedicated speaker embeddings, the end-to-end SLMs evaluated here process raw audio or discrete speech tokens directly. They lack an explicit "speaker verification module". This architecture explains their vulnerability to spoofing (Table 6) and high EER, as they must implicitly learn to disentangle speaker identity from semantic content without explicit supervision.
>
> **Table 13 of the revised manuscript. Speaker Verification Accuracy (ACC) and Equal Error Rate (EER) of Benchmarked SLMs.
> Performance was measured on a balanced binary task using audio from WenetSpeech.**
> | Model | Speaker Verification ACC (%) ↑ | Speaker Verification EER (%) ↓ |
> | --- | --- | --- |
> | Gemini-2.0-flash | 92.22 | 4.15 |
> | GLM4Voice | 74.40 | 12.80 |
> | Qwen2Audio | 68.51 | 15.93 |
> | Kimi-Audio | 60.90 | 21.05 |
> | Voxtral3B | 54.85 | 39.42 |
> | MiniCPM-2.6o | 51.19 | 48.17 |
> | Baichuan-Omni-1.5 | 50.83 | 48.73 |
> | Qwen2.5Omni | 50.12 | 49.13 |
>
> ### **(3) Fine-Tuning Dataset Justification (Response to Weakness 3)**
>
> >“Weakness: The construction of the 4000-hour mixed-task dataset for fine-tuning, including mixtures of tasks and proportions, is not fully justified. Ablation studies... are missing.”
>
> `Please refer to Line 1134-1138 on Page 22 of the revised manuscript.`
>
> Our construction of the 4,000-hour training set was driven by the need to balance new privacy skills with general capabilities.
>
> - **Justification:** We empirically found that a roughly **30% ratio of general-purpose data** (ASR, SER, etc., detailed in Table 12) was the "sweet spot".
> - **Ablation:** Our ablation study (Table 7, "Ours-ablation") demonstrates that removing this general data leads to catastrophic forgetting on standard benchmarks. The mixed-task approach is therefore essential for a viable SLM.

---

> ### Author Response · Authors · 2025-11-23
> **Reply to Reviewer m294 (Part 3/3)**
>
> ### **(4) Error Analysis and Multilingual Gaps (Response to Weakness 4, Questions 2 & 4)**
>
> >“Weakness: The underperformance on Chinese w.r.t. English is observed but is not sufficiently analyzed...”
>
> >“Question: Can the authors provide error analyses by secret categories and instruction phrasing...?”
>
> >“Question: Regarding the multilingual aspect, do the authors have hypotheses or insights on why models underperform for Chinese?”
>
> `Please refer to Line 369-374 on Page 7, Line 1461-1489 on Page 28 of the revised manuscript.`
>
> - **Error Analysis by Category:** We analyzed failure rates across the secret categories (from Table 1). The distribution reveals a distinct pattern between Tiers (**Figure 8 of the revised manuscript**):
>     - **Tiers 1 & 2 (Explicit Instructions):** Failures are distributed relatively evenly across categories. Interestingly, highly sensitive topics like *Personal Info* and *Illicit Actions* have slightly lower failure rates (~9%), suggesting base model safety alignment offers some residual protection.
>     - **Tier 3 (Proactive/Implicit):** Failures are heavily **content-dependent**. Models struggle most with nuanced categories like *Professional Aspirations* (18% of failures) and *Belief Conditions* (18%), while performing better on obvious secrets like *Personal Info*. This confirms that Tier 3 successfully tests social reasoning rather than just instruction following.
>
> **The source table of the Figure 8 of the revised manuscript. Distribution of privacy failures across secret categories for Tier 1, Tier 2, and Tier 3 tasks.**
> | Secret Category | Tier 1 | Tier 2 | Tier 3 |
> | --- | --- | --- | --- |
> |  Personal Info  |  9%  |  11%  |  8%  |
> |  Location Info  |  12%  |  13%  |  10%  |
> |  Academic Background  |  14%  |  12%  |  12%  |
> |  Interpersonal Secrets  |  13%  |  14%  |  16%  |
> |  Professional Aspirations |  14%  |  12%  |  18%  |
> |  Belief Conditions  |  13%  |  13%  |  18%  |
> |  Illicit Actions  |  9%  |  11%  |  5%  |
> |  Transient Secrets  |  16%  |  14%  |  13%  |
>
> - **Cross-Lingual Performance (Chinese vs. English):** We attribute the performance gap to two main factors:
>     1. **ASR Performance:** We used Kimi-Audio to transcribe the probe utterances from our benchmark. We found that the Word Error Rate (WER)/Character Error Rate (CER) for Chinese was **1.5-2x higher** than for English. These transcription errors propagate and degrade the downstream LLM's ability to reason about the semantic content.
>     2. **LLM Bias:** The underlying LLMs are predominantly English-centric, resulting in weaker commonsense reasoning and context tracking in Chinese, particularly for the subtle "implied privacy" tasks in Tier 3.
>
> Thank you once again for your constructive and valuable feedback. We hope our responses and the additional analyses have addressed your concerns.

---

> > ### Comment · Reviewer_m294 · 2025-11-26
> > **Response to the rebuttal**
> >
> > I thank the authors for the great work they did in addressing all the comments and questions.
> > All my questions were thoroughly addressed, supported by additional experimental results.
> > The previously noted weaknesses have been resolved. I recommend this paper for acceptance.

---

> > > ### Author Response · Authors · 2025-11-27
> > >
> > > Dear Reviewer m294,
> > >
> > > Thank you for your thoughtful review and positive feedback. We are pleased that our revisions have successfully addressed your concerns.

---

### Author Response · Authors · 2025-11-21

We sincerely thank all ACs and reviewers for their constructive feedback and high-level engagement with our work. We are encouraged that there is broad agreement that **interactional privacy for SLMs in shared, multi-user environments is both novel and important, and that VoxPrivacy fills a clear gap not addressed by existing speech, multi-speaker, or safety benchmarks.** Our goal in the rebuttal and revision was to (i) sharpen the practical motivation and threat model, (ii) strengthen the realism and validity of the benchmark with new human studies and real speech, and (iii) deepen the diagnostic analysis of where and why current SLMs fail.

### **Core contributions and what is new in the revision**

To the best of our knowledge, **VoxPrivacy is the first benchmark that directly evaluates interactional privacy in multi-speaker SLM dialogues**, with a **three-tier task structure** that spans explicit secrecy commands, speaker-verified access, and proactive privacy inference, plus a 4000 hour training set and a fine-tuned SLM baseline. In the revision, we substantially reinforce this contribution along three main axes, and **two of the reviewers (m294 and KeNS) felt we had fully addressed all of their concerns:**
1. **Realism of the threat model and motivation.**

    We clarify that VoxPrivacy targets **asynchronous, time-separated interactions on shared devices with memory**, not just concurrent multi-party chat. The three turns in our benchmark compress a realistic sequence: (T₀) user A discloses a secret, (T₁) sets an explicit or implicit privacy preference, (T₂) later user B queries the same assistant. We argue more clearly why **external permission systems (account logins, voice match, device locks) are insufficient and inadequate** in this setting: they either require impractical pre-enrollment of every possible speaker, or they enforce hard isolation between user histories, which breaks collaborative use cases (e.g., “What did Alex say about the meeting time?”). This strengthens the case that **speaker-aware, context-sensitive privacy reasoning inside the SLM is necessary, not a “nice-to-have”**.
2. **Validation of synthetic data via human studies and real recordings.**

    Multiple reviewers questioned whether synthetic, LLM-generated secrets and TTS audio truly reflect real-world privacy concerns and acoustic conditions. We therefore added two new validation components:

    - A **human perception study** where 5 bilingual annotators rated 200 randomly sampled “secrets” on a 5-point Likert scale. **92% of secrets received an average sensitivity rating ≥4**, showing that humans indeed treat these scenarios (including Tier-3 examples like “I’m worried about my medical results”) as information they expect an assistant not to reveal to others.
    - A **Real-VoxPrivacy evaluation** in which **18 bilingual volunteers recorded 586 utterances** (balanced across EN/ZH). The new Table 4 shows that **model rankings, the Tier2 to Tier3 difficulty gap, and the near-random behavior of most open-source SLMs all persist on real human speech**. In some cases, performance on real audio is even worse due to accents and noise, confirming that our synthetic benchmark is a **clean “best-case” upper bound rather than an overestimate of real-world difficulty**.

---

> ### Author Response · Authors · 2025-11-21
>
> 3. **Richer analysis of error patterns, speaker verification, cross-lingual gaps, and evaluation protocol.**
>     - We add **error analyses by secret category and instruction tier**, showing that Tiers 1–2 failures are relatively evenly spread (with slightly *lower* failure rates on universally sensitive categories like *Personal Info* and *Illicit Actions*), whereas Tier 3 errors concentrate on nuanced categories such as *Professional Aspirations* and *Belief Conditions*. This supports our claim that Tier 3 is genuinely testing **commonsense social reasoning**, not just more of the same instruction-following.
>     - We provide a **speaker verification analysis with Equal Error Rate (EER)** alongside accuracy, showing a strong negative correlation and highlighting that current SLMs, which lack explicit SV modules, are especially vulnerable to spoofing attacks. This addresses concerns about biometric features and gives a more conventional view of SV performance.
>     - We deepen the **cross-lingual analysis** for English vs Chinese: (i) ASR error rates for Chinese are roughly **1.5–2× higher** than for English, propagating noise into downstream reasoning; and (ii) model backbones remain English-centric, which especially harms subtle Tier3 privacy inference in Chinese. These findings, plus Real-VoxPrivacy, clarify that the EN–ZH gap is rooted in both front-end and language modeling asymmetries rather than an artifact of our benchmark design.
>     - We also refine our **evaluation terminology and validation**: we now refer to “LLM-based” vs “Human” evaluation (rather than “objective/subjective”), and we report that human annotators achieve **high agreement (high Fleiss’ κ)** and closely track the LLM-as-judge scores, confirming that the automatic metric is a reliable and reproducible proxy for human judgments.
>
> ### **Conclusion**
> We believe VoxPrivacy establishes a critical foundation for the safe deployment of SLMs. By releasing the benchmark (32h), the Real-VoxPrivacy validation set, the 4,000-hour training set, and the fine-tuned baselines, we aim to facilitate great progress in this vital area. We thank the reviewers again for helping us strengthen the paper significantly.

---

### Meta-Review · Area_Chair_FBR7 · 2026-01-01

**Summary:**

This paper proposed VoxPrivacy: A Benchmark for Evaluating Interactional Privacy of Speech Language Models.

This paper received mixed review scores (4,4,8,10). The authors provided detailed responses. Importantly, in addition to the synthetic dataset in the original submission, real-world data are provided during rebuttal. The “positive” reviewers are satisfied with the responses. Unfortunately, the “negative” reviewers did not engage in the discussion.

Both reviewers muR9 and G4Sy mentioned concerns on motivation of the paper. Reviewer muR9's concern can be (at least partially) resolved by the authors' clarification on asynchronous memory when interacting with a speech assistant. However, I am not sure if Reviewer G4Sy's comment on permission systems is fully addressed. In general, since this scenario is not extensively discussed before, and such a permission system does not exist yet, the authors' arguments look reasonable.

Because of the interests from reviewers, this paper is generally welcomed to the community. I appreciate the efforts on creating and open-sourcing benchmarks, and would encourage the authors to incorporate the reviewers’ feedback to improve the benchmark.

**Reviewer Concerns:**

- One of the main concerns from reviewers is that the dataset is synthetic. The concern is (at least partially) resolved by the additional real-world dataset provided during rebuttal.

- Reviewer muR9 and G4Sy also mentioned concerns on motivation of the paper. Reviewer muR9's concern can be (at least partially) resolved by the authors' clarification on asynchronous memory when interacting with a speech assistant. However, I am not sure if Reviewer G4Sy's comment on permission systems is fully addressed. In general, since this scenario is not extensive discussed before, and such a permission system does not exist yet, I am supportive of the authors' argument.

**Reviewer Scores:**

Reviewer m294 and KeNS are satisfied with the rebuttal, and would maintain their positive scores (8, 10).

Reviewer muR9's comment on motivation might be partially resolved by clarification on asynchronous memory, and might flip from borderline reject to borderline accept 4 -> 6.

Reviewer G4Sy's comment on motivation is not quite addressed, but might slightly increase score due to the additional real-world dataset. Possibly 4->6.

---

### Decision · Program_Chairs · 2026-01-26

Accept (Poster)